# An expanded toolkit for *Drosophila* gene tagging using synthesized homology donor constructs for CRISPR-mediated homologous recombination

**Oguz Kanca[1,2]\*, Jonathan Zirin[3], Yanhui Hu[3], Burak Tepe[1,2], Debdeep Dutta[1,2], Wen-Wen Lin[1,2], Liwen Ma[1,2], Ming Ge[1,2], Zhongyuan Zuo[1,2], Lu-Ping Liu[3], Robert W Levis[4], Norbert Perrimon[3,5], Hugo J Bellen[1,6]\***

[1]Department of Molecular and Human Genetics, Baylor College of Medicine, Houston, United States; [2]Duncan Neurological Research Institute, Texas Children Hospital, Houston, United States; [3]Department of Genetics, Harvard Medical School, Boston, United States; [4]Department of Embryology, Carnegie Institution for Science, Baltimore, United States; [5]Howard Hughes Medical Institute, Harvard Medical School, Boston, United States; [6]Department of Neuroscience, Baylor College of Medicine, Houston, United States

**\*For correspondence:**
kanca@bcm.edu (OK);
hbellen@bcm.edu (HJB)

**Competing interest:** The authors declare that no competing interests exist.

**Abstract** Previously, we described a large collection of *Drosophila* strains that each carry an artificial exon containing a *T2AGAL4* cassette inserted in an intron of a target gene based on CRISPR-mediated homologous recombination. These alleles permit numerous applications and have proven to be very useful. Initially, the homologous recombination-based donor constructs had long homology arms (>500 bps) to promote precise integration of large constructs (>5 kb). Recently, we showed that in vivo linearization of the donor constructs enables insertion of large artificial exons in introns using short homology arms (100–200 bps). Shorter homology arms make it feasible to commercially synthesize homology donors and minimize the cloning steps for donor construct generation. Unfortunately, about 58% of *Drosophila* genes lack a suitable coding intron for integration of artificial exons in all of the annotated isoforms. Here, we report the development of new set of constructs that allow the replacement of the coding region of genes that lack suitable introns with a *KozakGAL4* cassette, generating a knock-out/knock-in allele that expresses GAL4 similarly as the targeted gene. We also developed custom vector backbones to further facilitate and improve transgenesis. Synthesis of homology donor constructs in custom plasmid backbones that contain the target gene sgRNA obviates the need to inject a separate sgRNA plasmid and significantly increases the transgenesis efficiency. These upgrades will enable the targeting of nearly every fly gene, regardless of exon–intron structure, with a 70–80% success rate.

## Editor's evaluation

This manuscript is of general interest to *Drosophila* researchers who widely use the many tools generated by the Gene Disruption Project ( GDP). This is a valuable addition to this toolkit. The approach in this paper generates new vectors, which allow the rapid generation of hundred of gene-specific GAL4 lines using CRISPR technology. The approach taken by the authors has important implications outside the *Drosophila* community too.

## Introduction

The *Drosophila* Gene Disruption Project (GDP) aims to generate versatile genetic tools for most genes to facilitate the study of gene function in vivo and to create fly stocks for the community. The CRISPR-mediated integration cassette (CRIMIC) approach is a recent addition to the GDP to target fly genes. The CRIMIC strategy is based on integrating a Swappable Integration Cassette (SIC) containing an artificial exon encoding *attP-FRT-Splice Acceptor (SA)-T2AGAL4-polyA-3XP3EGFP-polyA-FRT-attP* (*T2AGAL4*). The SIC is integrated in an intron between two coding exons (coding intron) by CRISPR-mediated homologous recombination (*Lee et al., 2018*; *Gnerer et al., 2015*; *Diao et al., 2015*). The viral T2A sequence leads to the truncation of the nascent target gene polypeptide and reinitiation of translation of the downstream GAL4 as an independent protein. This cassette typically creates a strong loss of function allele of the targeted gene and expresses the yeast GAL4 transcription factor in a similar spatial and temporal pattern as the protein encoded by the targeted gene (*Lee et al., 2018*). These alleles can be used to: (1) determine the gene expression pattern; (2) study the effect of loss of function of the gene product; (3) replace the SIC through Recombinase Mediated Cassette Exchange (RMCE) (*Bateman et al., 2006*; *Venken et al., 2011*) with an artificial coding exon that encodes a fluorescent protein to assess protein subcellular localization (*Venken et al., 2011*) and identify interacting proteins; (4) express UAS-cDNAs of the targeted gene and its variants to assess rescue of the mutant phenotype and conduct structure/function studies (*Wangler et al., 2017*); (5) excise the insert with UAS-Flippase to confirm that removal of the insertion will revert the phenotype (*Lee et al., 2018*).

The introduction of an artificial exon is only feasible for genes that contain a suitable large coding intron, typically 100 nt or more (*Lee et al., 2018*). This requirement makes more than half of the genes inaccessible to strategies based on the use of artificial exons (*Supplementary file 1*). In addition, the genes that do not have a suitable intron are typically smaller in size than genes that contain a suitable intron, and usually have fewer previously isolated alleles than larger genes with a suitable intron (*Supplementary file 1*).

Here, we describe the development of a knock-out/knock-in strategy to replace the coding sequence of genes with a *Kozak sequence-GAL4-polyA-FRT-3XP3EGFP-polyA-FRT* (*KozakGAL4*) cassette to target genes that lack introns that are suitable for artificial exon knock-ins. The targeted gene is cut by Cas9 using two sgRNAs, one targeting the 5′ untranslated region (UTR) and the other the 3′ UTR. We targeted ~100 genes with this strategy and show that about 80% of the integrated KozakGAL4 cassettes lead to *UAS-mCherry* expression in the third instar larval brain, a ratio that is similar to what was observed for the *T2AGAL4* strategy (*Lee et al., 2018*).

We also improved the design of the homology donor constructs that can be used for integration of either *KozakGAL4* or *T2AGAL4* cassettes. We previously showed that short homology arms combined with the linearization of the donor plasmid DNA, in vivo results in integration of large constructs (~5 kb) through homologous recombination (*Kanca et al., 2019b*). This allows commercial DNA synthesis of the entire gene-specific portion of the donor plasmid, a cheaper and more efficient option than PCR amplification and cloning of each homology arm. To further extend this approach, we developed a method in which the DNA sequences directing the transcription of the target gene-specific sgRNA(s) are synthesized together in the same segment as the homology arms. This design eliminates the need to clone and inject a separate vector for the sgRNAs. We tested our new designs on ~200 genes and show that the upgrades result in a transgenesis efficacy of ~80%. The strategies that we introduce here allow targeting of nearly every gene in the fly genome, further streamline the generation of homology donor DNAs, increase efficiency as compared to previous strategies, and improve the rate of precise genome editing.

## Results and discussion
### The KozakGAL4 cassette

Integration of an artificial exon is only feasible for genes that have coding introns large enough to identify an sgRNA target site that is located at a sufficient distance from the preceding splice donor and the following splice acceptor site. Based on our experience, an intron should be larger than 100 nt to be suitable for integration of an artificial exon. An analysis of the *Drosophila* genome shows that 5787 out of 13,931 protein-coding genes have a sufficiently large coding intron that is shared among all the annotated splicing isoforms of the gene (*Supplementary file 1*). In our analysis, we focused on

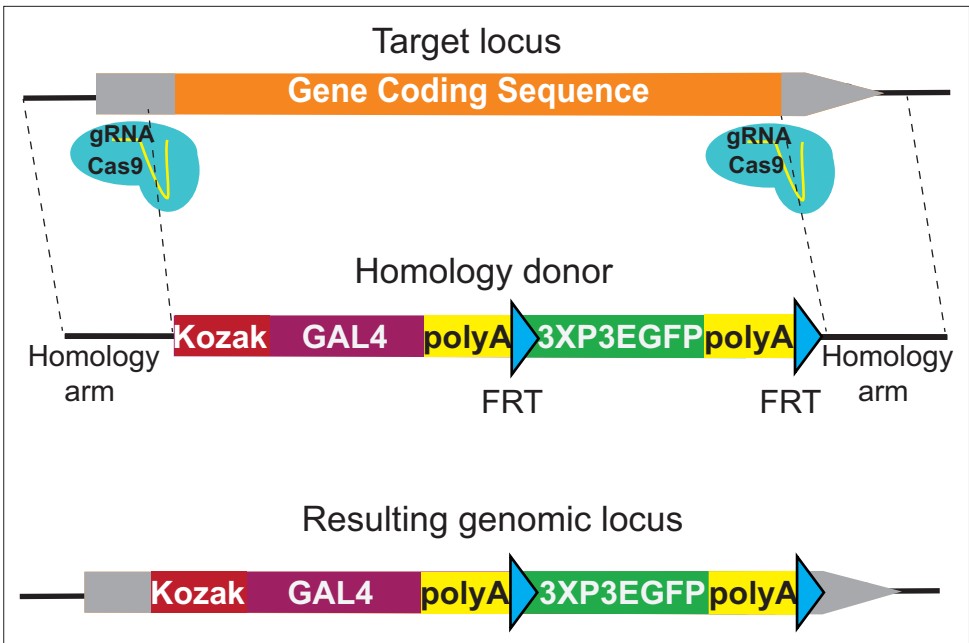

**Figure 1.** *KozakGAL4* strategy can be used to generate *GAL4* gene trap alleles.
Schematics of the *KozakGAL4* targeting. Gray boxes, UTRs; orange box, gene-coding region.

The online version of this article includes the following figure supplement(s) for figure 1:

**Figure supplement 1.** Alternative strategies to generate gene trap alleles in genes without suitable introns.

the suitable introns that target all the annotated transcripts for the GDP, to affect all possible isoforms of the targeted gene. This is a stringent criterion and individual laboratories that aim to generate isoform-specific alleles can use suitable introns that affect only some of the isoforms or generate multiple insertions for the same gene to cover all the annotated isoforms using our methodology. Using these criteria, 8144 genes lack suitable introns, making them less accessible for the *T2AGAL4* and other artificial exon-based strategies.

*Supplementary file 1* compares the 5787 genes with a suitable coding intron (as defined above) with the 8,144 genes lacking a suitable intron. Genes with a suitable intron typically have larger coding sequences (2051 nt vs 1173 nt) and have a larger number of previously isolated mutant alleles based on FlyBase data (10.4 vs 2.1) than genes without a suitable intron(s) (*Supplementary file 1*; *Larkin et al., 2021*). To integrate a *GAL4* cassette that can be used in a similar manner as the *T2AGAL4* insertions, we developed the *KozakGAL4* knock-in/knock-out strategy. The Kozak sequence is an optimal translation initiation site in eukaryotic mRNAs, and it is identified as (C/A) AA (C/A) AUG in *Drosophila* (*Cavener, 1987*; *Kozak, 1986*). We use CAAA as a Kozak sequence upstream of the start codon of *GAL4*. To replace the coding region of genes, we typically identify sgRNA target sites in the 5′ UTR and 3′ UTR (*Figure 1*). To retain possible gene expression regulation by the 5′ UTR, we select the upstream sgRNA target site that is closest to the start codon and that is not predicted to have off-target activity based on CRISPR Optimal Target Finder (*Gratz et al., 2014*). The location of the downstream sgRNA target site in the 3′ UTR is less stringent since the endogenous 3′ UTR is not included in the final transcript due to the *polyA* signal in the *KozakGAL4* cassette. The median 5′ and 3′ UTR lengths for *Drosophila* genes are 214 and 224 bps, respectively (*Chen et al., 2011*; *Jan et al., 2011*), which are typically large enough to identify putative sgRNA targets. In our experience the 5′ UTRs typically contain multiple sgRNA targets whereas 3′ UTRs contain few candidate sgRNA target sites due to their A/T rich nature. In cases where a suitable sgRNA target site cannot be found in the 3′ UTR, we target a site within the coding region, close to the stop codon, to minimize the coding region of the gene that remains. In cases where a suitable sgRNA site cannot be found within the 5′ UTR region, the search is expanded to the promoter region and the sequence between the sgRNA cut site and transcription start site is added to the homology region. In such cases, a single nucleotide substitution

to eliminate the PAM sequence is introduced in the homology donor construct, preventing cutting of the homology donor.

We also developed alternative strategies to target genes without suitable introns and for which no proper sgRNA site could be identified within the 5′ UTR or promoter. We generated the *SA-KozakGAL4-polyA-3XP3EGFP-polyA* cassette that can be introduced in an intron within the 5′ UTR (*Figure 1—figure supplement 1A*). Alternatively, for genes with small coding introns, we make two cuts: one within a coding intron just upstream of the SA of an exon; the second in the 3′ UTR. The excised sequence is then replaced with a *T2AGAL4* cassette (*Figure 1—figure supplement 1B*).

## KozakGAL4 alleles drive expression of UAS transgenes

The expression of the yeast GAL4 protein in *Drosophila* in a temporal/spatial pattern that mimics a *Drosophila* gene has been a key tool for functional genetics. There have been two main approaches to generate alleles that express GAL4 in the pattern of a *Drosophila* gene that are not based on the *T2AGAL4*-based strategies. The first one is based on an insertion of minimal promoter-*GAL4*-coding sequences in a transposon backbone. The strategy is called enhancer trapping and was based originally on *GFP* and *LacZ* rather than *GAL4* (*O'Kane and Gehring, 1987*; *Bellen et al., 1989*). Upon mobilization of the transposon, lines are established where the GAL4 expression pattern is of interest (*Brand and Perrimon, 1993*; *Lukacsovich et al., 2001*; *Hacker et al., 2003*; *Metaxakis et al., 2005*; *Bellen et al., 2011*; *Gohl et al., 2011*). Given that *GAL4* cassettes are inserted in the genome by transposable elements in this strategy, they are not always optimally placed to report the full expression pattern of the gene (*Spradling et al., 1995*; *Mayer et al., 2013*), but they have been used extensively as many reflect the expression pattern of a nearby gene (*Wilson et al., 1989*). Many however are not mutagenic (*Spradling et al., 1999*). The second strategy to generate alleles that may express GAL4 in the expression pattern of a gene is to clone a 500 bp to 5 kb region upstream of the promoter of the gene upstream of the GAL4-coding sequences and inserting the transgene in the genome. There are large collections of these enhancer-*GAL4* alleles and most aim to report the expression of enhancer fragments rather than reporting the expression pattern of the gene from which they are derived (*Jenett et al., 2012*; *Manning et al., 2012*; *Pfeiffer et al., 2008*). Additionally, two ends-out homologous recombination based knock-in strategies were developed to target *rabGTPase* genes (*Chan et al., 2011*; *Jin et al., 2012*). These methods were based on recombineering to generate homology donor constructs that either replace the entire coding sequence of the gene with *GAL4*-coding sequence or knock-in a *GAL4*-coding sequence to replace the ATG codon-containing exon of the targeted *rabGTPase*. These *GAL4* knock-in alleles of *rabGTPases* were used to determine and compare the phenotypes and expression domains of all 29 *rabGTPase* genes in the genome. These approaches can now be complemented and expanded with the *KozakGAL4* approach that should incorporate all or most upstream regulatory information and to generate a null allele of the targeted gene. The latter greatly facilitates rescue experiments using UAS-cDNA transgenes.

To assess the potential of the *KozakGAL4* strategy, we targeted 109 genes to date and successfully replaced the coding region of 82 genes with the *KozakGAL4* cassette (*Supplementary file 2*). We crossed 57 of these with *UAS-CD8mCherry* transgenic flies to determine the GAL4 expression of the targeted gene in the brain of wandering third instar larvae. Our previous findings, using *T2AGAL4* alleles have shown that ~80% of all *T2AGAL4* alleles lead to specific expression in third instar larval brains (*Lee et al., 2018*). Similarly, with *KozakGAL4* alleles, we detected specific GAL4 expression for about 80% (46/57) of the genes (*Figure 2A*). Although *KozakGAL4* targeted genes are typically small, limiting the possible regulatory information in the coding region, it is possible that the removal of coding and small introns as well as some UTR sequences may affect the regulatory input. We therefore tested whether a few *KozakGAL4* alleles drive the expression of the *UAS-CD8mCherry* in a similar pattern as the targeted gene. We selected a *KozakGAL4* allele that drives expression of the reporter in a restricted group of cells in the third instar larval brain (*CG10939^KozakGAL4*) and analyzed the single-cell RNA sequencing (scRNAseq) data for the third instar larval brain to determine cell clusters that express the targeted gene (*Ravenscroft et al., 2020*). We then used the same scRNAseq dataset to determine other genes expressed in overlapping clusters and that were previously targeted with *T2AGAL4*. The *KozakGAL4*-driven *UAS-CD8mCherry* reporter is expressed in a very similar expression pattern as the *T2AGAL4*-driven reporter expression of the genes that we identified through scRNAseq. The other genes expressed in the overlapping cluster according to scRNAseq such as *serp*, *verm*, and

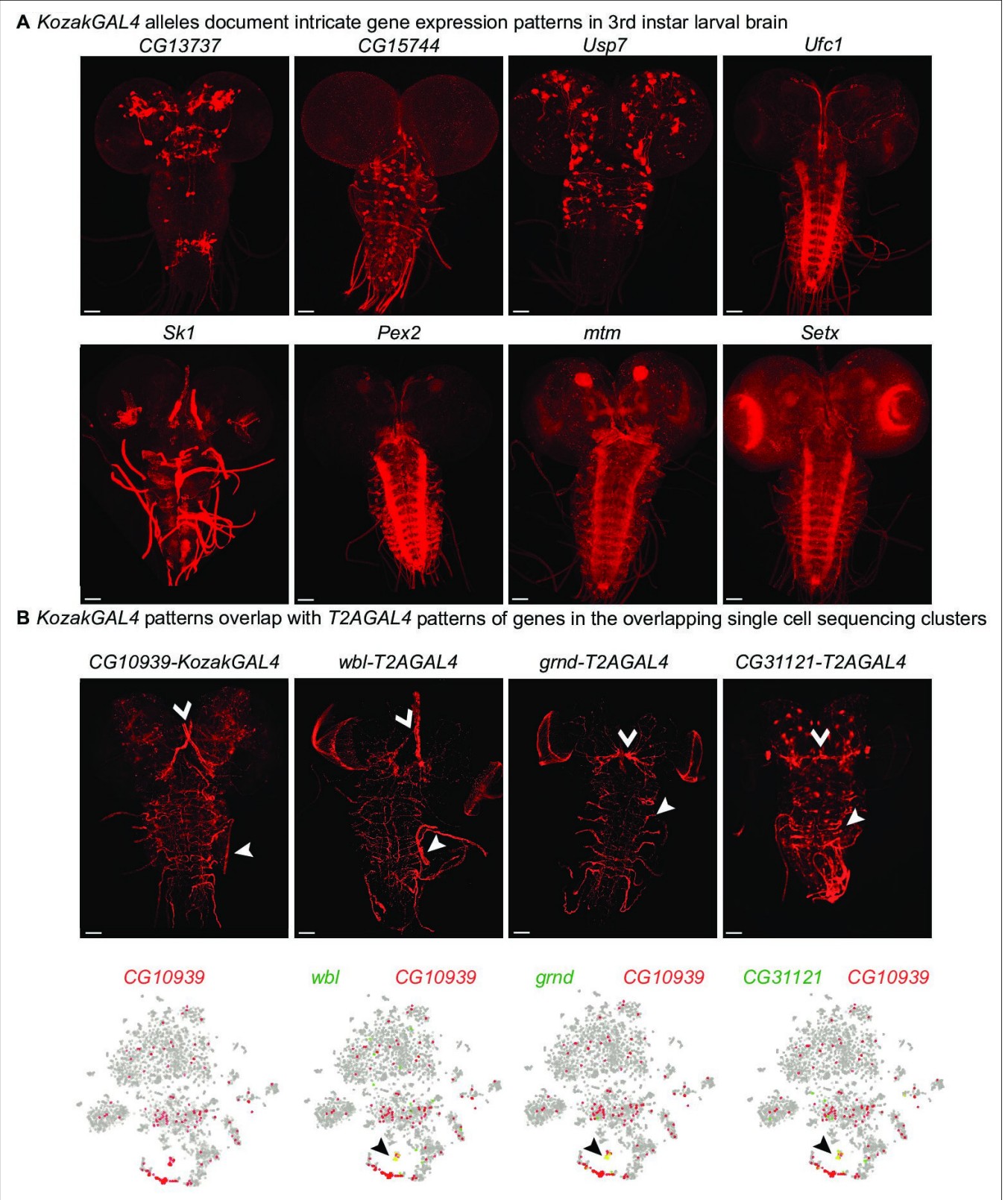

**Figure 2.** *KozakGAL4* alleles document intricate gene expression patterns in third instar larval brains. (**A**) Examples of third instar larval brain gene expression patterns obtained by crossing *KozakGAL4* allele of indicated genes with *UAS-CD8mCherry* flies. (**B**) The imaging results of reporter expression generated with *KozakGAL4* allele were compared to the expression pattern of genes that are expressed in similar cells by analysis of single-

*Figure 2 continued on next page*

*Figure 2 continued*

cell sequencing data imaged using *T2AGAL4* alleles. Images are taken by crossing the *GAL4* alleles with *UAS-CD8mCherry*. Arrowheads point to the shared expression pattern. Scale bar is 50µm.

The online version of this article includes the following figure supplement(s) for figure 2:

**Figure supplement 1.** Identification of genes expressed in similar cells to the *KozakGAL4* alleles expressed in restricted patterns.

**Figure supplement 2.** Comparison of mRNA expression of *GAL4* and targeted genes mRNA for *KozakGAL4* alleles.

*emp* suggest that this cluster corresponds to tracheal cells, which is in line with the observed expression pattern based on imaging (*Figure 2B*, *Figure 2—figure supplement 1B*; *Luschnig et al., 2006*; *Lee et al., 2018*). We also performed the analysis for three additional *KozakGAL4* alleles that lead to reporter expression in restricted groups of cells (*CG3770*, *CG10947*, and *CG15093*). Comparison of the expression patterns of the other tested *KozakGAL4* targeted genes and *T2AGAL4* targeted genes that are expressed in overlapping cell groups showed again overlapping expression patterns based on imaging (*CG3770^{KozakGAL4}-Debcl^{T2AGAL4}*; *CG10947^{KozakGAL4}-PK2-R1^{T2AGAL4}*; *CG15093^{KozakGAL4}-pippin^{T2AGAL4}*; *Figure 2—figure supplement 1A*). Hence, the use of scRNAseq data can provide an independent means of verification of accuracy of the observed reporter expression patterns for the lines tested.

Next, we compared the RNA expression patterns of the *GAL4* mRNA and the targeted gene mRNA for eight genes targeted by *KozakGAL4* in flies heterozygous for a *KozakGAL4* allele. We opted to compare mRNA expression patterns rather than comparing the signal obtained by *KozakGAL4/UAS-Fluorescent Protein* to the protein expression pattern of the targeted gene because there are no available antibodies for vast majority of the genes we targeted. Moreover, if the antibodies would be available, and even if the antibodies were completely specific, the staining pattern would likely be different from the *GAL4-UAS* reporter expression pattern due to the subcellular localization of the gene product being different from the subcellular localization of the reporter protein. Finally, the *GAL4-UAS* system amplifies the signal compared to the expression of the gene product. *Lee et al., 2018* noted the greater level of expression provided by the *T2AGAL4* system when they used RMCE to convert the same MiMIC lines to *EGFP* protein trap alleles or *T2AGAL4* gene trap alleles. The signals that they detected in larval or adult brains using these alleles often looked qualitatively different as well. Comparing the expression pattern of the targeted gene product to the *KozakGAL4-UAS* reporter gene signal would suffer from the same issue.

For a more direct comparison of the spatial/temporal expression of the *KozakGAL4* allele and the targeted gene, we compared their RNA expression patterns by in situ hybridization. We employed smiFISH (single molecule Fluorescent In-Situ Hybridization) in third instar larval brains for eight genes (*Calvo et al., 2021*; *Yang et al., 2017*). We crossed the *KozakGAL4* alleles of these genes (*Setx*, *CG7943*, *CG8202*, *CG8778*, *CG15093*, *IntS11*, *ZnT49B*, and *Vps29*) to *yw* flies with wild-type alleles of these genes and performed costaining of the *GAL4* mRNA, expressed by the *KozakGAL4* allele, and the targeted gene mRNA, expressed from the wild-type allele of the gene. In seven cases, where we could detect the mRNA expression of the gene product reliably (*Setx*, *CG7943*, *CG8202*, *CG8778*, *CG15093*, *IntS11*, and *ZnT49B*), four showed clear overlap between the gene product and GAL4 mRNAs (*Setx*, *CG8202*, *CG10593*, and *IntS11*, *Figure 2—figure supplement 2*). Three additional cases, *CG7943*, *CG8778*, and *ZnT49B* had very low signal-to-noise ratio and are inconclusive. There were no cases where we saw clear target gene product staining and no Gal4 overlap. Taken together these data suggest that the knock-ins are expressed in the correct gene pattern but given the low sample size we do not draw any strong conclusion as to how consistent this is across the genome.

Next, we determined if UAS-human or fly cDNAs could rescue the phenotype associated with 11 gene deletions caused by the *KozakGAL4* knock-ins. UAS-fly or human cDNA rescued the lethality of *KozakGAL4* allele for *Pngl* (Clement Chow, personal communication), *Tom70*, *CG8320*, *CG16787*, and *IntS11* and behavioral phenotypes for *Wdr37*, suggesting that the *KozakGAL4* is expressed where the targeted gene product is required for the gene function. For *Pex2*, *Pex16*, *Fitm*, *PIG-A*, and *CG34293* the expression of orthologous human cDNA did not rescue the associated phenotypes.

In summary, *KozakGAL4* offers a means to disrupt gene function while expressing GAL4 in the expression domain of the targeted genes with the caveat that possible exonic and intronic regulatory information would be removed. This approach allows us to tag the remainder of the genes that do not contain a suitable coding intron for the *T2AGAL4* strategy which corresponds to 58% of all the genes.

## New vector backbones for synthesis of homology donor constructs that are also templates for sgRNA expression

We previously showed that linearizing the homology donor constructs in vivo allows for integration of large constructs in the genome through CRISPR-mediated homologous recombination even using short homology arms (*Kanca et al., 2019b*). This approach makes inexpensive commercial synthesis of homology donor intermediates feasible. The intermediate vectors can be used for a single step directional cloning of the SIC in the homology donor intermediate vector. This greatly facilitates the generation of homology donor vectors which previously required four-way ligations with large homology arms. Moreover, this eliminates cloning failures (~20–30%) and troubleshooting associated constructs with large homology arms (*Kanca et al., 2019b*). The resulting homology donor vectors with short homology arms were previously injected together with two vectors that express two sgRNAs (pCFD3, *Port et al., 2014*) in embryos that express Cas9 in their germline. The first sgRNA, which we refer to as sgRNA1, targets the homology donor vector backbone to linearize the homology donor and does not have a target in the *Drosophila* genome (*Garcia-Marques et al., 2019*). The second sgRNA vector

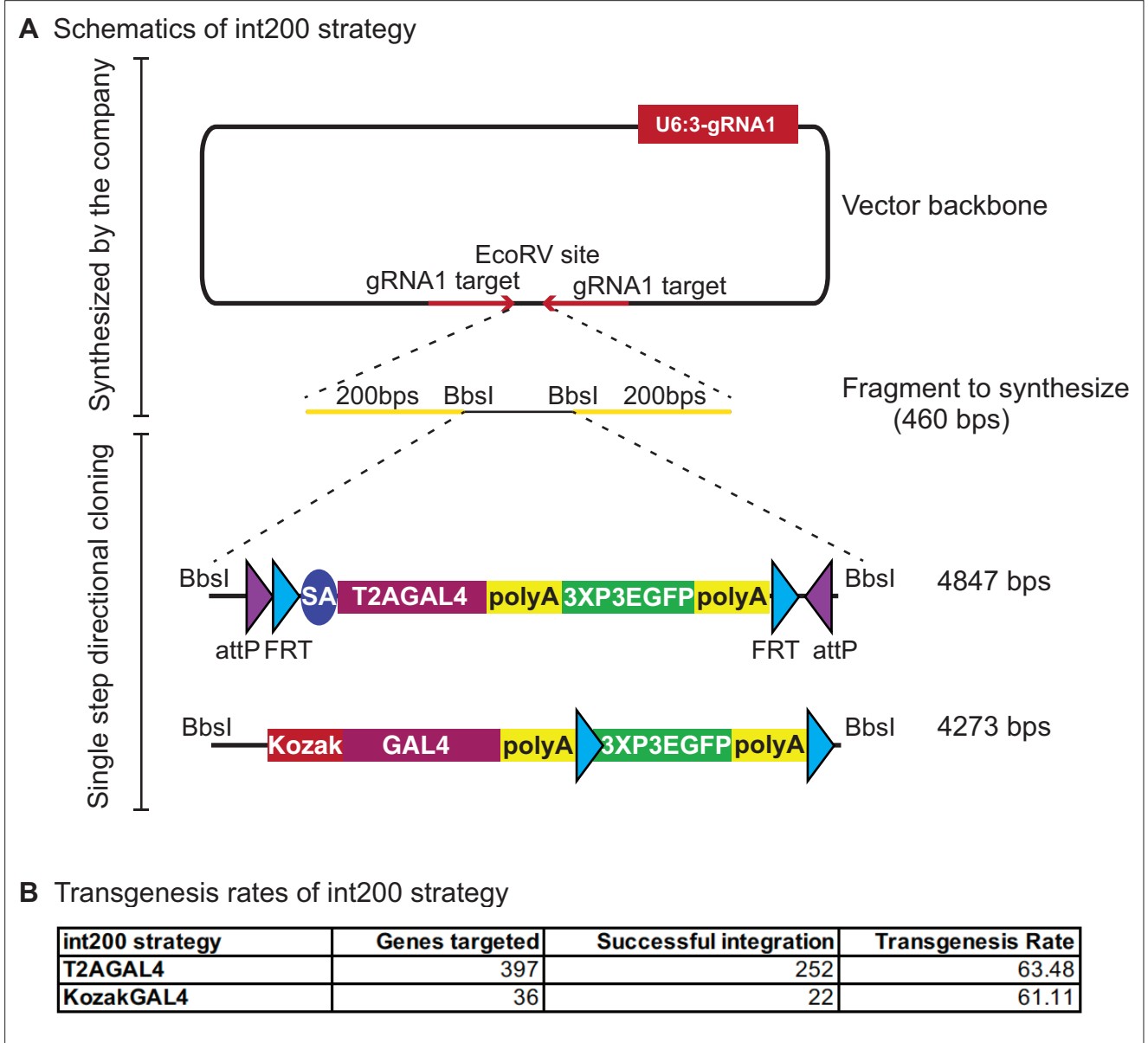

**Figure 3.** int200 strategy results in similar transgenesis success rates as the long homology arms CRISPR-mediated integration cassettes (CRIMICs). (**A**) Schematics of the int200 strategy. (**B**) Transgenesis data using int200_T2AGAL4 or int200_KozakGAL4 strategies.

expresses the sgRNA to target the gene and introduce the double strand DNA break that serves as a substrate for homologous recombination. In *Kanca et al., 2019b*, we demonstrated that injection of these constructs resulted in transgenesis efficiencies of about 60%.

We developed new approaches to increase the transgenesis efficiency of the custom DNA backbones, decrease the workload, and to simplify the generation of homology donor constructs. The first custom vector backbone that we tested has the U6-3::sgRNA1 sequence in the vector backbone and sgRNA1 targets, on either side of the EcoRV site where the synthesized fragments are directionally integrated by Gibson Assembly by the synthesis company (vector backbone named pUC57_Kan_gw_OK, design named int200, *Figure 3A*). With this design, the homology donor vector intermediates that are commercially synthesized contain the sgRNA1-coding sequence, obviating the need to coinject one of the sgRNA vectors. Having the sgRNA1-coding region in the backbone also helps with in vivo linearization of the homology donor since the homology donor construct and the sgRNA1 are delivered together in a single vector. The int200 design also removes the sgRNA1 target sites from the synthesized region as they are present in the vector. This allows increasing the homology arm length to 200 bps without increasing the cost of synthesis.

We tested the int200 design for 397 genes with *T2AGAL4* cassettes and 36 genes with *KozakGAL4* cassettes. For each construct, we injected 400–600 embryos that express Cas9 in the germline. For inserting the *KozakGAL4* cassette, the two gene-specific sgRNAs were cloned into pCFD5 (*Port and Bullock, 2016*). For both *T2AGAL4* and *KozakGAL4* insertions, the int200 homology donor plasmid was coinjected with the plasmid that encodes the target-specific sgRNA (pCFD3 for the former and pCFD5 for the latter case). We successfully integrated *T2AGAL4* cassette in 252 genes (~65% success rate) and replaced the coding region of 22 genes with *KozakGAL4* (~61% success rate) (*Figure 3B*). PCR verification of each of the two flanks of each insertion site was performed by using a gene-specific PCR primer outside the homology region pointing toward the insert and a construct specific PCR primer. For 88% of the *T2AGAL4* inserts we obtained PCR verification on both sides of the insert and for the remaining 12% we obtained PCR products on one side of the construct. For 91% of *KozakGAL4* inserts we obtained amplicons on both sides of the insert. For the inserts with a single PCR verification, we sequenced the amplicon to ensure the insert is in the proper locus. Hence, the overall transgenesis success rate of the int200 method is 63% (*Figure 3B*). This is very similar to the injection success rate of homologous recombination using large (0.5–1 kb) homology arms (1165 insertions in 1,84 targeted genes, *Lee et al., 2018*) but leads to very significant reductions in labor and cost. Additionally, int200 facilitates the cloning of homology donor constructs and eliminates cloning failures which reduce the overall successful targeting rate using large homology arms to ~50% (successful cloning of 80% constructs that are injected with 65% transgenesis success rate). In summary, the int200 method provides a ~30% gain in overall efficiency (from 50% to 65%).

To further optimize the custom vector backbone, we repositioned the U6-3::sgRNA1 and added a partial tRNA construct directly upstream of the EcoRV site where the synthesized fragments are inserted (*Figure 4A*). The placement of the partial tRNA allows seamless integration of the remainder of the tRNA and gene-specific sgRNA sequence to the synthesized fragment (vector named pUC57_Kan_gw_OK2 and design named gRNA_int200 for *T2AGAL4* constructs and named 2XgRNA_int200 for *KozakGAL4* constructs). Hence, two sgRNAs (or three sgRNAs for *KozakGAL4*) can be produced from the single injected plasmid. One of the sgRNA1 target sites is added to the synthesized fragment before the start of the homology arm and the other sgRNA1 target site is added to the backbone just downstream of the EcoRV site where the synthesized fragment is directionally inserted. This design obviates the need to clone a separate sgRNA vector to target the genomic locus. It also ensures simultaneous delivery of all the components of the homologous recombination reaction as they are delivered on a single plasmid. We have targeted 127 genes with gRNA_int200_T2AGAL4 donor plasmids (*Supplementary file 2*) and successfully inserted the *T2AGAL4* cassette in 95 genes (~75% success rate, *Figure 4B*). We also tested whether genes for which the tagging failed using the int200 strategy (*Figure 3*) could be targeted with the gRNA_int200_T2AGAL4 using the same gene-specific sgRNA and homology arms. For three out of four genes tested, use of the gRNA_int200 strategy resulted in successful integration of the *T2AGAL4* cassette. These data show that incorporating all the sgRNAs in the donor vector improves the transgenesis efficiency.

For the *KozakGAL4* constructs, we inserted a second tRNA sequence after the first synthesized sgRNA and added the second gene-specific sgRNA sequence to the synthesis reaction. We targeted

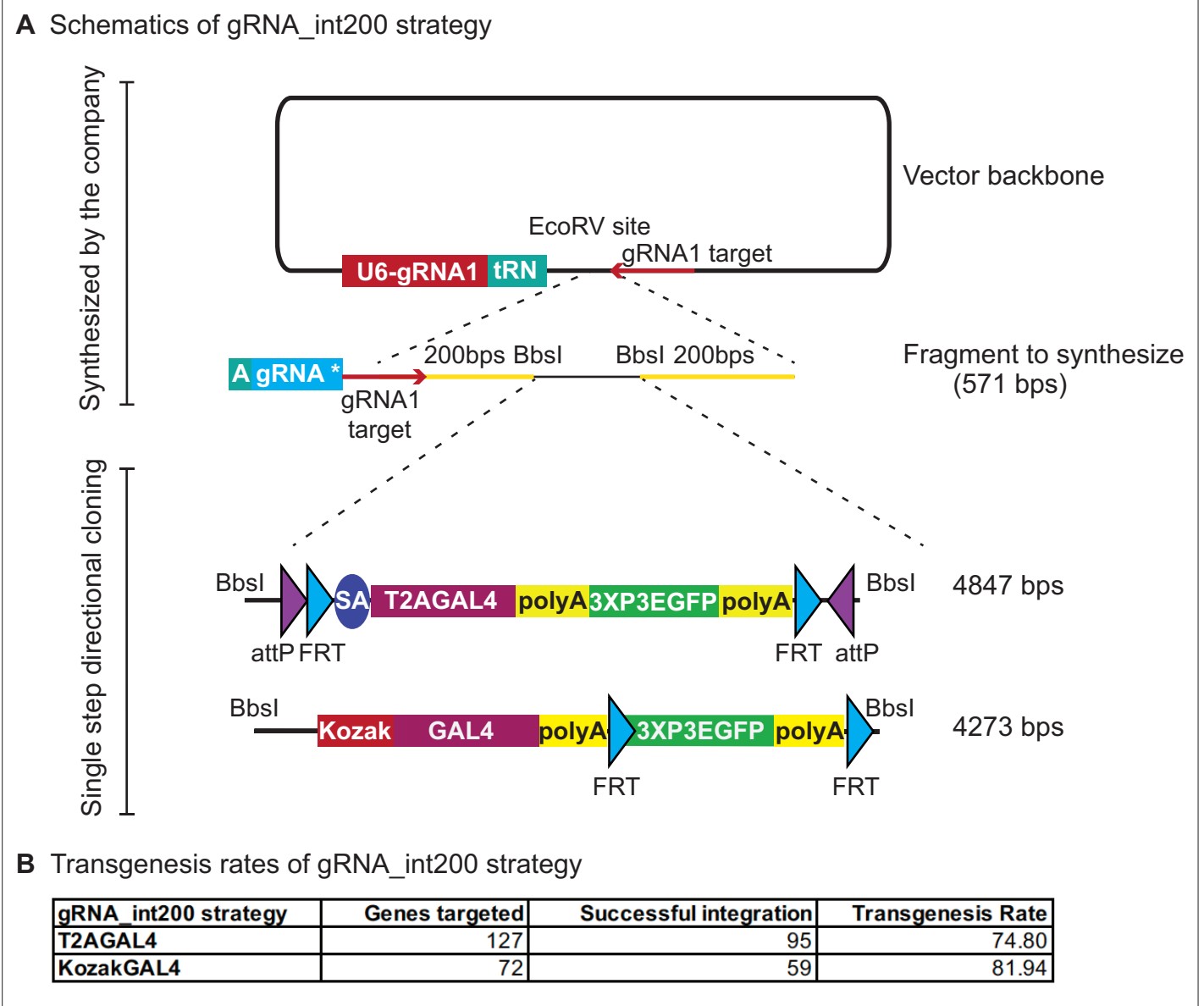

**Figure 4.** gRNA_int200 strategy increases the transgenesis success rates. (**A**) Schematics of the gRNA_int200 strategy. (**B**) Transgenesis data using gRNA_int200_T2AGAL4 or 2XgRNA_int200_KozakGAL4 strategies.

72 genes with the 2XgRNA_int200_KozakGAL4 cassette and successfully inserted *KozakGAL4* in 59 genes (~82% success rate, *Figure 4B*; *Supplementary file 2*). We tested whether genes for which the tagging failed using the int200_KozakGAL4 strategy (*Figure 3*) could be properly targeted with the 2XgRNA_int200_KozakGAL4 using the same gene-specific sgRNAs and homology arms sequences and again observed that for three out of four tested genes, the 2XgRNA_int200_KozakGAL4 strategy was successful. In summary, the gRNA_int200 design increases the transgenesis rate and streamlines the creation of *T2AGAL4* and *KozakGAL4* constructs by obviating the need to generate a separate sgRNA expression plasmid and ensuring codelivery of all components for homologous recombination. In summary, the gRNA_int200 allows a 78% transgenesis success rate, or an additional 20% when compared to the int200 approach (increase from 65% to 78%).

## Use of 2XgRNA_int200 intermediate vectors for GFP tagging

We have previously shown that integrating a *SA-linker-EGFP-FlAsH-StrepII-3xTEVcs-3xFLAG-linker-SD* (*SA-GFP-SD*) in coding introns of genes is an efficient approach to tag proteins with GFP (*Venken*

*et al., 2011*; *Nagarkar-Jaiswal et al., 2015b*; *Nagarkar-Jaiswal et al., 2017*; *Li-Kroeger et al., 2018*). We typically generate these alleles through RMCE of existing MiMIC SICs and we have shown that they are functional in 72% of tested genes (*Nagarkar-Jaiswal et al., 2015b*; *Nagarkar-Jaiswal et al., 2015a*). A major factor that affects the functionality of the GFP protein trap is the insertion position. In cases where the artificial exon encoding the protein trap is inserted in a coding intron that bisects a predicted functional protein domain, the resulting protein trap is often not functional. Hence, it would be highly desirable to have another efficient approach to tag proteins encoded by genes that have no intron, small introns or no suitable MiMICs in any preselected position in the protein structure.

We tested the use of synthesized homology donor intermediate vectors to replace the coding sequence of a gene with the gene-coding sequence fused to GFP at different locations. We selected the *Wdr37* gene as it has a small intron (*Kanca et al., 2019a*). We amplified *Wdr37* sequences from the genome by PCR and used NEB HiFi DNA assembly to generate homology donor constructs where a sfGFP tag is integrated at the N terminus, C terminus, or internally (*Figure 5*, *Figure 5—figure supplements 1–3* for schematics of HiFi assembly). The 3XP3 DsRed flanked by PiggyBac transposase inverted repeats is integrated after the 3' UTR and serves as the transformation marker that can be excised precisely using the PiggyBac transposase (http://flycrispr.molbio.wisc.edu/; *Bier et al., 2018*; *Figure 5A*). The assembled sequences are subcloned in the synthesized homology donor intermediate. Injection of the homology donor plasmids into embryos expressing Cas9 in their germline resulted in positive transgenics in each case. A western blot of the resulting protein trap alleles using anti-GFP antibody detected bands at the expected length for the tagged protein in each case. However, the internally tagged allele is less abundant, underlining that the placement of the sfGFP tag can affect protein stability (*Figure 5B*). Hence, the strategy to replace the whole coding region with a GFP tagged coding region allows tagging almost any gene in any position in the coding sequence.

In summary, we developed a *KozakGAL4* strategy to target the genes that do not have a suitable intron and a set of novel custom vector backbones to facilitate homology donor construct production and increase transgenesis rate. The methods we developed are versatile and can be modified to generate *GAL4* gene traps or GFP protein fusions of the targeted genes. Finally, the methodology we describe should be easy to implement in any other model organisms to facilitate generation of gene trap and protein trap alleles.

# Materials and methods
## Generation of homology donor constructs
Templates for ordering the int200 and gRNA_int200 constructs and detailed explanation of construct design and cloning steps can be found in *Supplementary file 3*. Homology donor intermediate vectors were ordered for production from Genewiz ('ValueGene' option) in pUC57 Kan_gw_OK (for int200 strategy) or pUC57 Kan_gw_OK2 (for gRNA_int200 strategy) vector backbone at 4 µg production scale. The lyophilized vectors were resuspended in 53 µl of ddH$_2$O. 1 µl was used for Golden Gate assembly with 290 ng of pM37 vector of reading frame phase corresponding to the targeted intron (for *T2GAL4*, *Lee et al., 2018*) or 265 ng of pM37_KozakGAL4 vector (for *KozakGAL4*). The Golden Gate Assembly reaction was set in 200 µl PCR tubes (ThermoScientific AB2000) with 2.5 µl 10× T4 DNA ligase buffer (NEB B0202S), 0.5 µl T4 DNA ligase (NEB M0202L), 1 µl restriction enzyme (BbsI_HF or BsaI_HFv2 NEB R3559L and R3733L, respectively), 1 µl of SIC (pM37_T2AGAL4 or pM37_KozakGAL4 at 290 or 265 ng/µl, respectively), 19 µl of dH$_2$O, and 1 µl of homology donor construct. For cloning multiple constructs in parallel, master mixes were prepared including all the components except for the homology donor intermediate vector. The reactions were incubated in a Thermocycler (cycle 30 times between 37°C 5 min, 16°C 5 min, then 65°C 20 min, 8°C hold). An additional digestion step was done to remove self-ligating plasmid backbones by adding 19.5 µl dH$_2$O, 5 µl 10× CutSmart buffer (NEB B7204S) and 0.5 µl BbsI or BsaI_HFv2 (the enzyme used for the cloning reaction). The reaction product was transformed in DH5α competent cells and plated on Kanamycin + LB plates.

## Fly injections
Int200-T2AGAL4 and int200-KozakGAL4 constructs were injected at 250 ng/µl along with 100 ng/µl gene-specific gRNA(s) cloned in pCFD3 or pCFD5, respectively (*Port et al., 2014*; *Port and Bullock, 2016*). Injections were performed as described in *Lee et al., 2018*. 400–600 embryos from *y¹w*<sup>*</sup>; iso18;*

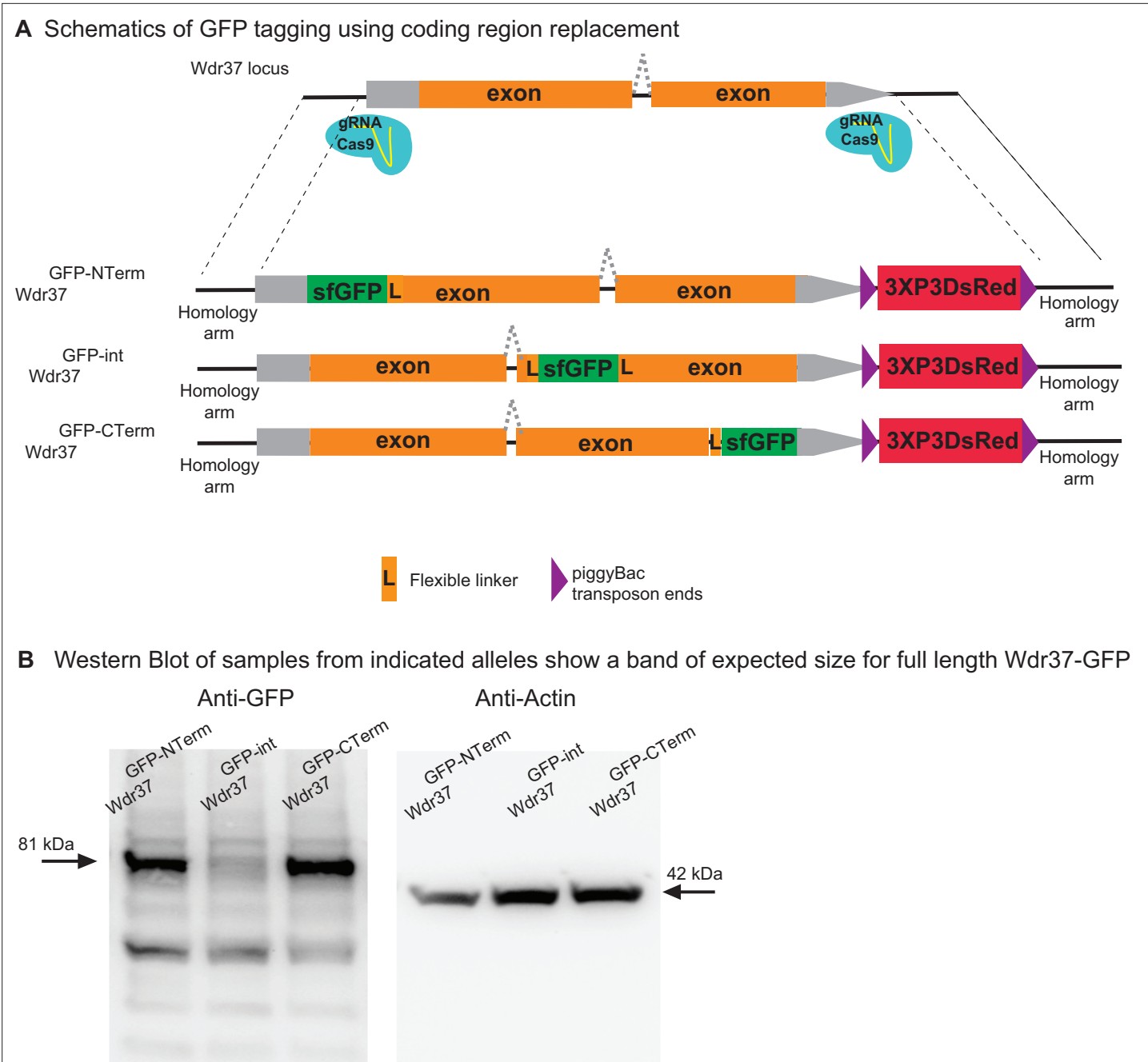

**Figure 5.** 2XgRNA_int200 strategy can be used to tag any gene at any coding region to generate protein trap alleles. (**A**) Schematics of the targeting constructs to integrate sfGFP protein tag at an N-terminal, internal, or C-terminal location in *Wdr37* gene locus. (**B**) Western blot analysis from adult flies show full-length protein in all protein trap alleles with the arrow indicating the 81 kDa band that is the length predicted for the Wdr37 protein fused to sfGFP.

The online version of this article includes the following source data and figure supplement(s) for figure 5:

**Source data 1.** Source data for the western blots from *Figure 5*.

**Figure supplement 1.** Schematic of Wdr37 GFP-NTerm protein trap allele donor construct.

**Figure supplement 2.** Schematic of Wdr37 GFP-int protein trap allele donor construct.

**Figure supplement 3.** Schematic of Wdr37 GFP-CTerm protein trap allele donor construct.

attP2(y+){nos-Cas9(v+)} for genes on the second or fourth chromosome and $y^1w^*$ iso6;; attP2(y+){nos-Cas9(v+)} for genes on the X chromosome and $y^1w^*$; attP40(y+){nos-Cas9(v+)}; iso5 (**Kondo and Ueda, 2013**) for genes on the third chromosome per genotype were injected. Whole genome sequencing BAM files of isogenized lines can be found at https://zenodo.org/record/1341241. Resulting G0 males and females were crossed individually to $y^1w^*$ flies as single fly crosses for 3XP3-EGFP detection. Positive lines were balanced, and stocks were established. Up to five independent lines were generated per construct per gene. The list of generated alleles can be found in **Supplementary file 2**. The sequences of homology arms and sgRNA(s) as well as the results of PCR validation and imaging on third instar larval brain are available at https://flypush.research.bcm.edu/pscreen/crimic/crimic.php. The stocks are deposited in the Bloomington *Drosophila* Stock Center (BDSC) on a regular basis. The stocks are available from the Bellen lab until they are deposited and established in the BDSC.

## PCR validation

PCR primers that flank the integration site were designed for each targeted gene. These primers were used in combination with SIC-specific primers that bind 5′ of the inserted cassette in reverse orientation and 3′ of the insert in forward orientation (pointing outwards from the insert cassette). SIC-specific primer sequences can be found in **Supplementary file 4**. 200–800 nt amplicons were amplified from genomic DNA from individual insertion lines through single fly PCR (**Gloor et al., 1993**) using OneTaq PCR master mix (NEB #M0271L). PCR conditions were 95°C for 30 s, 95°C 30 s, 58°C 30 s, 68°C 1 min for 34 cycles, and 68°C 5 min.

## Confocal imaging of transgenic larval brains

Dissection and imaging were performed following the protocols in **Lee et al., 2018**. In brief, fluorescence-positive third instar larvae were collected in 1× phosphate-buffered saline (PBS) solution and then cut in half and inverted to expose the brain. Brains were transferred into 1.5 ml centrifuge tubes and fixed in 4% Paraformaldehyde (PFA) in 1× PBS buffer for 20 min. Brains were then washed for 10 min three times in 0.2% PBS Triton X-100 (PBST). Finally, samples were mounted on glass slides with 8 µl of VectaShield (VectorLabs #H-1000) and imaged at 20× zoom with a Nikon W1 dual laser spinning-disc confocal microscope.

## smiFISH assay for dual detection of *KozakGAL4* and targeted gene mRNA

smiFISH probes for the indicated genes and *GAL4* are designed using Biosearch Technologies Stellaris RNA FISH probe designer tool (https://www.biosearchtech.com/support/tools/design-software/stellaris-probe-designer). Flap sequence 'CCTCCTAAGTTTCGAGCTGGACTCAGTG' were added at the 5′ end of each probe generating 48mer oligonucleotides. For *GAL4*, *Setx*, *ZnT49B*, *CG8208*, *CG7943*, and *IntS11*, 48 probes were designed. mRNAs of *Vps29*, *CG8778*, and *CG15093* were not long enough to design 48 probes. We designed 30 probes for *Vps29*, 32 probes for *CG8778*, and 34 probes for *CG15093*. Probe sequences can be found in **Supplementary file 5**. Probes were ordered from Sigma-Aldrich at 0.025 µM scale normalized to 100 µM per well in plates. X-FLAP (CACTGAGTCCAGCTCGAAACTTAGGAGG) oligos tagged with Cy3 or Cy5 were also ordered from Sigma-Aldrich at 0.025 µM synthesis scale. Gene-specific oligos were pooled and annealed to X-FLAP oligos as described in **Calvo et al., 2021** at 20 µM scale.

*KozakGAL4/+* third instar larvae were dissected and stained as specified in **Yang et al., 2017**. Samples were incubated with gene-specific probes labeled with Cy5 and *GAL4*-specific probes labeled with Cy3 by annealing to respective oligos. 1 µM final concentration of each probe were used for incubating the third instar larval brains for 12 hr. *yw* larval brains were used as a negative control for *GAL4*-specific probes.

## Analysis of single-cell sequencing data

To identify genes with expression profiles that overlap with expression of genes replaced with *KozakGAL4* sequences, we queried the data from third instar larval CNS scRNAseq data described in **Ravenscroft et al., 2020**. The data (http://scope.aertslab.org/#/Larval_Brain/*/welcome) were imported into Seurat (version 4.0.1). Cells expressing the selected genes, for which the KozakGAL4 allele was generated (e.g., *CG3770*, *CG10939*, *CG10947*, and *CG15093*), were identified using

WhichCells function and genes enriched in these cells were identified using FindMarkers with default parameters. A list of the top 10 genes that were minimally expressed outside the expression domain of genes with KozakGAL4 alleles was generated. We then selected genes from the list for which T2AGAL4 were generated and compared the expression profiles using available images.

## Western blots

Flies were homogenized using Cell Lysis Buffer (25 mM Tris–HCl pH 7.5, 100 mM NaCl, 1 mM Ethylenediaminetetraacetic acid (EDTA), 1% Triton X-100, 1× liquid protease inhibitor [Gen DEPOT], 0.1 M Dithiothreitol (DTT)). The supernatant was collected after centrifugation at 13,000 rpm for 10 min at 4°C (Eppendorf 5424R with rotor Eppendorf FA-45-24-11). The supernatant was mixed with Laemmli Buffer containing β-mercaptoethanol and heated at 95°C for 10 min. Subsequently, the samples were loaded in 4–20% gradient polyacrylamide gels (BioRad Mini-PROTEAN TGX). Following electrophoresis, proteins were transferred onto a polyvinylidene difluoride membrane (Immobilon, Sigma). The membrane was blocked using skimmed milk and treated with the primary antibody for overnight. The following antibodies were used in the present study: rabbit anti-GFP (1:1000) (Thermo Fisher Scientific, #A-11122), mouse anti-Actin (1:5000) (EMD Millipore, #MAB1501). Horseradish peroxidase-conjugated secondary antibody was used to detect the respective primary antibody. Blots were imaged on a BioRad ChemiDocMP.

## Cloning of *Wdr37-KIGFP* constructs

The fragments that position the GFP tag to the selected sites were PCR amplified from genomic DNA, sfGFP was amplified from pBS_SA_sfGFP_SD (*Kanca et al., 2019a*) and scarless DSred from pScarlessHD-DsRed pScarlessHD-DsRed was a gift from Kate O'Connor-Giles (Addgene plasmid # 64703; http://www.addgene.org/64703/; RRID:Addgene_64703). The fragments were used together with homology donor intermediate for *Wdr37* gene used for generating *KozakGAL4* allele (CR70111) cut with BbsI-HF to assemble NEB-HiFi DNA assembly following manufacturer's instructions. Schematics of HiFi assembly can be found in *Figure 5—figure supplements 1–3*. Primer sequences to clone the homology donor constructs can be found at *Supplementary file 4*.

## Acknowledgements

We thank the Bloomington *Drosophila* Stock Center (BDSC) for acquisition and distribution of the fly stocks generated by the GDP. We thank both the BDSC and FlyBase for curation of the data associated with these stocks, facilitating their usage by the research community. We thank the Developmental Studies Hybridoma Bank for antibodies. We thank Stephanie Mohr for critical reading of the manuscript. We thank Megan Cooper for technical help with smiFISH experiments. We thank Yuchun He, Ying Fang, Minhua Huang, Zhihua Wang, Yaping Yu, Junyan Fang, Ruifang Zhang, and Lily Wang for generating and maintaining MiMIC/CRIMIC *T2AGAL4* fly stocks. We also thank Dr. Karen Schulze for her help in establishing the CRIMIC database and her previous work for the GDP. Confocal microscopy was performed in the BCM IDDRC Neurovisualization Core, supported by the NICHD (U54HD083092). HJB received support from NIH R01GM067858, R24OD031447, R24OD022005, U54NS093793 and the Huffington Foundation. OK received support from NIH R24 OD031447. JZ, YHu, and NP received support from NIGMS (GM067761 and GM084947). NP is an investigator of the Howard Hughes Medical Institute. RWL was supported by the Carnegie Institution for Science.

## Additional information

### Funding

| Funder | Grant reference number | Author |
|---|---|---|
| National Institute of General Medical Sciences | R01GM067858 | Hugo J Bellen |

| Funder | Grant reference number | Author |
|---|---|---|
| Office of Research Infrastructure Programs, National Institutes of Health | R24OD031447 | Oguz Kanca Hugo J Bellen |
| National Institute of Neurological Disorders and Stroke | U54NS093793 | Hugo J Bellen |
| Huffington Foundation | | Hugo J Bellen |
| National Institute of General Medical Sciences | GM067761 | Jonathan Zirin Yanhui Hu Norbert Perrimon |
| National Institute of General Medical Sciences | GM084947 | Jonathan Zirin Yanhui Hu Norbert Perrimon |
| Howard Hughes Medical Institute | | Norbert Perrimon |
| Carnegie Institution for Science | | Robert W Levis |

The funders had no role in study design, data collection, and interpretation, or the decision to submit the work for publication.

## Author contributions

Oguz Kanca, Conceptualization, Data curation, Formal analysis, Funding acquisition, Investigation, Methodology, Project administration, Resources, Supervision, Validation, Visualization, Writing – original draft, Writing – review and editing; Jonathan Zirin, Conceptualization, Data curation, Investigation, Methodology, Project administration, Resources, Validation, Writing – review and editing; Yanhui Hu, Data curation, Formal analysis, Investigation, Methodology, Software, Supervision, Validation, Writing – review and editing; Burak Tepe, Conceptualization, Formal analysis, Investigation, Methodology, Validation, Visualization, Writing – review and editing; Debdeep Dutta, Investigation, Methodology, Validation, Visualization, Writing – review and editing; Wen-Wen Lin, Data curation, Investigation, Methodology, Project administration, Supervision; Liwen Ma, Methodology, Validation, Visualization; Ming Ge, Investigation, Methodology, Validation, Visualization; Zhongyuan Zuo, Data curation, Methodology, Project administration, Visualization; Lu-Ping Liu, Methodology; Robert W Levis, Data curation, Formal analysis, Investigation, Project administration, Supervision, Validation, Writing – review and editing; Norbert Perrimon, Conceptualization, Funding acquisition, Project administration, Resources, Supervision; Hugo J Bellen, Funding acquisition, Project administration, Resources, Supervision, Writing – review and editing

## Author ORCIDs

Oguz Kanca (ID) http://orcid.org/0000-0001-5438-0879
Robert W Levis (ID) http://orcid.org/0000-0003-3453-2390
Norbert Perrimon (ID) http://orcid.org/0000-0001-7542-472X
Hugo J Bellen (ID) http://orcid.org/0000-0001-5992-5989

## Decision letter and Author response

Decision letter https://doi.org/10.7554/eLife.76077.sa1
Author response https://doi.org/10.7554/eLife.76077.sa2

# Additional files

## Supplementary files

• Supplementary file 1. Comparison of genes with or without suitable introns for *T2AGAL4* strategy. Comparison of the CDS length of the 5787 protein-coding genes with an intron suitable for the T2AGAL4 strategy with the 8144 genes lacking a suitable intron, because the intron is either absent, <100 nt, or is not present in all annotated transcript isoforms. The left column of the table

lists categories of CDS size or the number of alleles annotated in FlyBase (classical and transposon insertions). The number of genes in each category is given for genes with introns that are suitable for the T2AGAL4 strategy (middle column) and for genes lacking a suitable intron (right column). These data for the number of annotated alleles and the CDS size are presented in graphical format below the table.

• Supplementary file 2. List of the 428 alleles generated in this study. The genes for which we generated alleles are grouped under headers corresponding to the strategies used to generate the alleles.

• Supplementary file 3. Detailed design and cloning protocol. Step-by-step instructions and diagrams depicting construct design and cloning steps. Construct sequences and links to the digital construct sequence files are included in the protocol steps.

• Supplementary file 4. Primer sequences. Sequences of the primers used for PCR verification of alleles and generation of GFP knock-in donor constructs.

• Supplementary file 5. smiFISH probes. Sequences of the probes used in smiFISH experiment.

• Transparent reporting form

### Data availability
Source data for the graphs in Supplementary file 1 are included above the graphs in Supplementary file 1. The list of all the generated alleles and the method used can be found in Supplementary file 2.

The following previously published datasets were used:

| Author(s) | Year | Dataset title | Dataset URL | Database and Identifier |
|---|---|---|---|---|
| Ravenscroft TA, Janssens J, Lee PT, Tepe B, Marcogliese PC, Makhzami S, Holmes T, Aerts S, Bellen HJ | 2020 | 3rd instar brain scRNAseq data | https://www.ncbi.nlm.nih.gov/geo/query/acc.cgi?acc=GSE157202 | NCBI Gene Expression Omnibus, GSE157202 |
| Lee PT, Zirin J, Kanca O, Lin WW, Schulze KL, Li-Kroeger D, Tao R, Devereaux C, Hu Y, Chung V, Fang Y, He Y, Pan H, Ge M, Zuo Z, Housden BE, Mohr SE, Yamamoto S, Levis RW, Spradling AC, Perrimon N, Bellen HJ | 2018 | WGS of isogenized injection lines | https://zenodo.org/record/1341241 | 1341241, 1341241 |

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
