## [Editor Report]

This manuscript is of general interest to *Drosophila* researchers who widely use the many tools generated by the Gene Disruption Project ( GDP). This is a valuable addition to this toolkit. The approach in this paper generates new vectors, which allow the rapid generation of hundred of gene-specific GAL4 lines using CRISPR technology. The approach taken by the authors has important implications outside the *Drosophila* community too.

---

## [Decision Letter]

**Decision letter after peer review:**

Thank you for submitting your article "An expanded toolkit for *Drosophila* gene tagging using synthesized homology donor constructs for CRISPR mediated homologous recombination" for consideration by *eLife*. Your article has been reviewed by 2 peer reviewers, and the evaluation has been overseen by a Reviewing Editor and K VijayRaghavan as the Senior Editor. The following individual involved in the review of your submission has agreed to reveal their identity: Benjamin H White (Reviewer #1).

The reviewers have discussed their reviews with one another, and the Reviewing Editor has drafted this to help you prepare a revised submission, upon which the manuscript should be ready for acceptance,

Essential revisions:

Please see the two reviewers' reports below. An important point raised relates to improving the data on the fidelity of the KozakGal4 method. This could require more experiments, but the work will be on even stronger ground if addressed. At the very least, consistency (or inconsistency) with known patterns of tissue expression in published databases should be established for a representative subset of genes. Validation by some direct method, such as immunostaining where antibodies are available or in situ hybridization, would be highly desirable. Given that the authors have successfully targeted over 150 genes with KozakGal4 lines, they could probably find antibodies and/or make in situ probes for testing a subset of them, if they don't already have such data. Providing the reader with a better sense of the expression fidelity achievable using the method would be useful.

*Reviewer #1 (Recommendations for the authors):*

p.8 As noted in the general comments, the presentation on the fidelity of expression is confusing and needs to be changed. Part of the confusion as it is currently written stems from the fact that the name of the KozakGal4 line initially discussed (KozakGal4-CG10939) is not named in the text.

p. 9: The sentence beginning "The resulting new homology donor vectors were previously…" is very confusing, since it is not clear what "new" refers to, especially in the context of being "previously" injected. It seems like maybe some innovation since Kanca, 2019a is being referred to, but this doesn't seem to be the case. It would be helpful if this were rephrased so that it is clear that only the older technology is being described.

p. 10: the "success rates" referred to seem to indicate successful transgenesis (i.e. incorporation of the selection marker) rather than recovery of the desired insertion product (since PCR failed to verify correct insertion on both sides of the gene for some 12% of cases). Is this correct? It isn't clear from the text, and would be useful if the authors were explicit on this point.

p. 11: The increased success rate reported for gRNA-int200 seems somewhat surprising. Why might incorporating the sgRNAs into the same vector prove so much more effective--especially when several hundreds of embryos are being injected for each construct? Was the frequency of transformants for any given construct increased when the gRNA-int200 approach was used? One can understand going from a few transformants recovered to many transformants, but going from no transformants to transformants in three-quarters of cases seems surprising. What do the authors speculate is the cause?

p. 12: Direct genomic insertion of tags into genes of interest has become fairly commonplace using CRISPR-based methods. Is there a particular benefit of the author's approach?

Figure 1A: for clarity "avg" should be replaced by "avg length".

Figure 2B: why are there two arrowheads in the images?

Supplementary Table 1: Is there any particular reason to reproduce the left-side table, which has already been presented in Figure 1A?

*Reviewer #2 (Recommendations for the authors):*

Most importantly, the highlight of the paper that makes it suitable for publication in *eLife*, is the simple and universal gRNA_int200 strategy for HDR-mediated CRISPR-based gene targeting. All the other method improvements described in the manuscript become obsolete as they can be replaced by the sgRNA_int strategy. Therefore, the paper should focus and emphasize much more – or alternatively even culminate – in the detailed presentation of this method. -Maybe the renaming of the strategy may be helpful as we expect it to become a standard in the field. While the material and methods section are sufficiently described, the method would deserve much better visual representation by generating an extended Figure 4 that would guide the reader step by step through the use and application of the vector backbone.

In the main text the authors write that the synthesized fragments are inserted in a directional fashion using EcoRV, which is misleading as EcoRV is a blunt cutter. If we understand correctly the orientation of the synthesized fragment in Figure 3 does actually not matter. What matters there is the orientation of the Gal4 cassettes with respect to the 200bps homology arms and that is indeed directed by the BbsI cloning.

The situation appears to be different with the gRNA_int200 strategy in Figure 4. Again, mentioning directional insertion together with EcoRV is misleading. Further, the scheme in the first part (A) is confusing. The synthesized fragment of 571 bps starts with a "AgRNA*" part, but the dotted line does not include this part. Yet, as we understand this "AgRNA*" is part of the synthesized fragment that needs to be inserted into the backbone. The supplementary methods with more detailed cloning instructions supports the notion that the gene-specific gRNA 2 + gRNA scaffold is indeed part of the synthesized fragment. We therefore ask to correct and adapt the Figure accordingly. In addition, we suggest to extend the main text and to highlight in more detail the strategy as well as the relevance of the method for the field. Finally, we suggest that the authors add a detailed paragraph in which they guide the readers through an exemplary cloning of an insert into the sgRNA_int200 backbone. Because even after repeated reading, it was not apparent how a custom template should be ordered from gene synthesis providers to get a seamless restoration of the tRNA sequence. A very good example for a similar user manual has been provided by Port and Bullock (2016) where they describe in great detail the crucial steps for the successful use of pCFD5. This paper has been cited by the authors.

---

## [Author Response]

Reviewer #1 (Recommendations for the authors):p.8 As noted in the general comments, the presentation on the fidelity of expression is confusing and needs to be changed. Part of the confusion as it is currently written stems from the fact that the name of the KozakGal4 line initially discussed (KozakGal4-CG10939) is not named in the text.

We agree with the reviewer that part of the difficulty in explanation of the analysis arises from now naming the genes used in the analysis. In revision, we included the gene names of all the analyzed KozakGAL4-T2AGAL4 groups in the text to facilitate the interpretation of the analysis.

p. 9: The sentence beginning "The resulting new homology donor vectors were previously…" is very confusing, since it is not clear what "new" refers to, especially in the context of being "previously" injected. It seems like maybe some innovation since Kanca, 2019a is being referred to, but this doesn't seem to be the case. It would be helpful if this were rephrased so that it is clear that only the older technology is being described.

We changed the “The resulting new homology donor vectors” with “The resulting homology donor vectors with short homology arms” for better clarification.

p. 10: the "success rates" referred to seem to indicate successful transgenesis (i.e. incorporation of the selection marker) rather than recovery of the desired insertion product (since PCR failed to verify correct insertion on both sides of the gene for some 12% of cases). Is this correct? It isn't clear from the text, and would be useful if the authors were explicit on this point.

The success rates reflect the cases where the landing site of the construct is PCR verified on at least one side for the CRIMIC insertions and both sides on KozakGAL4 insertions. For CRIMICs where PCR verification is done only for one side, we sequence the PCR product.

p. 11: The increased success rate reported for gRNA-int200 seems somewhat surprising. Why might incorporating the sgRNAs into the same vector prove so much more effective--especially when several hundreds of embryos are being injected for each construct? Was the frequency of transformants for any given construct increased when the gRNA-int200 approach was used? One can understand going from a few transformants recovered to many transformants, but going from no transformants to transformants in three-quarters of cases seems surprising. What do the authors speculate is the cause?

We believe that injecting a single plasmid that contains all the components of the homologous recombination ensures co-delivery and simultaneous delivery of all the components to the germ cells. This may explain the increased transgenesis rate.

p. 12: Direct genomic insertion of tags into genes of interest has become fairly commonplace using CRISPR-based methods. Is there a particular benefit of the author's approach?

Our approach allows the use of single homology donor intermediate synthesized by a company to clone the gene coding sequence tagged at any desired location in the coding region. In addition, the same intermediate vector can be used to generate a Knock-out allele, Knock-out/Knock-in allele such as KozakGAL4 cassette or any other variant of the targeted gene. We believe that this approach streamlines the generation of tagged alleles and variant alleles, especially if multiple tag locations/variants are to be tested.

Figure 1A: for clarity "avg" should be replaced by "avg length".

We did change avg to avg length and moved figure 1A to the supplementary table 1 where the data is graphed.

Figure 2B: why are there two arrowheads in the images?

We wanted to attract attention to two regions where the expression patterns are especially similar.

Supplementary Table 1: Is there any particular reason to reproduce the left-side table, which has already been presented in Figure 1A?

We removed the table from Figure 1A and combined all the bioinformatic analysis in Supplementary table 1.

Reviewer #2 (Recommendations for the authors):Most importantly, the highlight of the paper that makes it suitable for publication in eLife, is the simple and universal gRNA_int200 strategy for HDR-mediated CRISPR-based gene targeting. All the other method improvements described in the manuscript become obsolete as they can be replaced by the sgRNA_int strategy. Therefore, the paper should focus and emphasize much more – or alternatively even culminate – in the detailed presentation of this method. -Maybe the renaming of the strategy may be helpful as we expect it to become a standard in the field. While the material and methods section are sufficiently described, the method would deserve much better visual representation by generating an extended Figure 4 that would guide the reader step by step through the use and application of the vector backbone.

We thank the reviewer for their suggestion. We included a detailed design and cloning protocol as a supplementary text to help with the adoption of the method by other laboratories. The visuals to help explain the method are included in that section.

In the main text the authors write that the synthesized fragments are inserted in a directional fashion using EcoRV, which is misleading as EcoRV is a blunt cutter. If we understand correctly the orientation of the synthesized fragment in Figure 3 does actually not matter. What matters there is the orientation of the Gal4 cassettes with respect to the 200bps homology arms and that is indeed directed by the BbsI cloning.The situation appears to be different with the gRNA_int200 strategy in Figure 4. Again, mentioning directional insertion together with EcoRV is misleading. Further, the scheme in the first part (A) is confusing. The synthesized fragment of 571 bps starts with a "AgRNA*" part, but the dotted line does not include this part. Yet, as we understand this "AgRNA*" is part of the synthesized fragment that needs to be inserted into the backbone. The supplementary methods with more detailed cloning instructions supports the notion that the gene-specific gRNA 2 + gRNA scaffold is indeed part of the synthesized fragment. We therefore ask to correct and adapt the Figure accordingly. In addition, we suggest to extend the main text and to highlight in more detail the strategy as well as the relevance of the method for the field. Finally, we suggest that the authors add a detailed paragraph in which they guide the readers through an exemplary cloning of an insert into the sgRNA_int200 backbone. Because even after repeated reading, it was not apparent how a custom template should be ordered from gene synthesis providers to get a seamless restoration of the tRNA sequence. A very good example for a similar user manual has been provided by Port and Bullock (2016) where they describe in great detail the crucial steps for the successful use of pCFD5. This paper has been cited by the authors.

We revised the detailed cloning protocol document to help reader understand the steps. The company generates the inserts in EcorV site directionally by using Gibson Assembly of synthesized fragments instead of conventional cloning. We included this information in the text to avoid confusion for the reader.